# FROM LLMS TO LRMS: RETHINKING PRUNING FOR REASONING-CENTRIC MODELS

## ABSTRACT

Model pruning is a widely-used technique to reduce the significant computational cost of large language models (LLMs). However, existing research suffers from two key limitations: (1) pruning is typically evaluated post-hoc on datasets unrelated to the original training corpus, leaving it unclear if the model's general capabilities are preserved; and (2) it has focused almost exclusively on standard instruction-following models (**LLM-instruct**). The recent rise of reasoning-augmented models (**LLM-think**), which generate explicit chain-of-thought steps, presents an unstudied challenge for established pruning methods due to their substantially different generation patterns. In this work, we conduct the first systematic investigation of pruning across both LLM-instruct and LLM-think families. We introduce a rigorous experimental framework that leverages the models' original training corpora for both pruning calibration and post-pruning recovery, enabling a faithful assessment of performance preservation than prior work. Across a comprehensive suite of static and dynamic pruning methods evaluated on 17 diverse tasks, we find that the effectiveness of pruning strategies differs significantly between the two model families. Our results reveal that techniques optimized for concise instruction-following do not seamlessly transfer to preserving complex, multi-step reasoning. This work provides critical insights and practical guidelines for efficiently compressing the next generation of reasoning-augmented LLMs.

## 1 INTRODUCTION

Large language models (LLMs) (Touvron et al., 2023; Jiang et al., 2023; Naveed et al., 2024) have rapidly transformed natural language processing, with their success driven primarily by strong **instruction-following capabilities**. By learning to understand and follow user instructions, LLMs can perform a wide range of tasks such as translation (Zhu et al., 2024; Xu et al., 2024; Pang et al., 2024) and dialogue (Abbasian et al., 2024; Liu et al., 2024; Guan et al., 2025) without the need to fine-tune a separate model for each task. This flexibility is made possible by large-scale pretraining and fine-tuning, which equip LLMs with broad generalization abilities (Kaplan et al., 2020). However, scaling also brings enormous computational costs, creating challenges for training (OpenAI et al., 2024; Lin et al., 2024), deployment (DeepSeek-AI et al., 2025b), and real-world usage on resource-limited platforms (Zhao et al., 2025).

To address these challenges, **pruning** has become one of the most widely studied efficiency techniques. By removing redundant parameters, attention heads, or entire layers (Sun et al., 2024; Ma et al., 2023; Men et al., 2024), pruning reduces both model size and inference cost while preserving much of the original performance. Existing work has largely focused on two strategies: **depth pruning**, which accelerates inference by removing layers (e.g., ShortGPT (Men et al., 2024), Shortened LLaMA (Kim et al., 2024)); and **width pruning**, which increases throughput by shrinking hidden dimensions (e.g., LLM-Pruner (Ma et al., 2023), SliceGPT (Ashkboos et al., 2024)). Together, these methods form a mature toolkit for improving efficiency in instruction-following LLMs.

Despite promising advances, most prior studies apply pruning in a post-hoc manner, typically using datasets unrelated to the original training corpus. C4 (Raffel et al., 2020) is unanimously used to compute calibration metrics for pruning, whereas Alpaca (Taori et al., 2023) is used for post-fine-tuning. Recent work (Williams & Aletras, 2023; Bandari et al., 2024) shows that downstream task performance is highly sensitive to the choice of calibration data. This leaves an important gap: *it*

*remains unclear whether pruning truly preserves a model's native broad capabilities, or merely adapts it to narrow downstream tasks.*

Meanwhile, the LLM landscape is evolving. The dominant paradigm is shifting from models that follow instructions to models that can also perform explicit reasoning (Xu et al., 2025; DeepSeek-AI et al., 2025a). [1] Unlike LLM-instruct models (Yang et al., 2024) that directly map prompts to responses, LLM-think models produce step-by-step reasoning traces before generating final outputs (Wei et al., 2022). This paradigm substantially improves performance on complex tasks but also yields excessively long generations, often spanning thousands of tokens (Chen et al., 2025; Yang et al., 2025a). Despite these differences, almost all existing pruning work has focused exclusively on LLM-instruct, leaving it unclear whether strategies designed for standard models can transfer effectively to reasoning-augmented ones. This gap motivates a key question: *does pruning require new strategies to remain effective in LLM-think models, or can existing approaches generalize?*

In this work, we revisit pruning through the lens of these two LLM families, leveraging settings where both models and their training data are fully accessible. For LLM-instruct, we adopt the open-sourced Tulu language model(Lambert et al., 2024), along with its complete instruction-following fine-tuning corpus. For LLM-think, we construct our own model by fine-tuning LLM-instruct on the OpenThoughts dataset (Guha et al., 2025), which aggregates diverse reasoning-focused corpora. This setup allows us to systematically test pruning methods while using the original training datasets both as calibration sets for pruning and as recovery data for post-fine-tuning. Unlike previous studies, this enables us to directly measure *whether pruning can maintain the full capabilities of both instruction-following and reasoning models when recovery is performed under their native training distributions*.

We conduct a comprehensive study across static depth pruning, static width pruning, and dynamic depth pruning, evaluating their impact on both LLM-instruct and LLM-think. Our experiments span **17 diverse tasks**, covering classification, code generation, mathematics, and open-ended reasoning. From this analysis, we derive several key insights and practical recommendations for pruning in the era of reasoning-augmented LLMs.

Our contributions are threefold:

1. We reframe pruning in the context of two major LLM families (LLM-instruct and LLM-think), highlighting the unique challenges posed by reasoning-augmented models.

2. We establish an experimental framework leveraging open training corpora, enabling pruning and recovery under the same data distributions used to train the original models.

3. Through extensive experiments, we show how pruning affects instruction following and reasoning, and identify which strategies best preserve performance across cases.

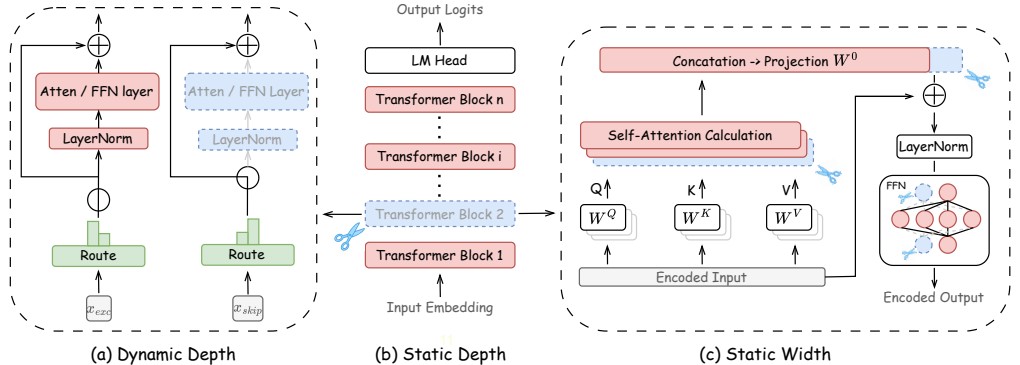

Figure 1: Overview of the three structured pruning strategies. Static depth pruning removes entire layers, static width pruning reduces hidden dimensions (neurons or attention heads), and dynamic depth pruning adaptively skips layers, attention blocks, or MLP modules depending on the input.

---

[1] For clarity, we refer to standard instruction-following LLMs as **LLM-instruct** and reasoning-augmented LLMs that output intermediate reasoning chains before answers as **LLM-think**.

## 2 BACKGROUND

### 2.1 LLM-INSTRUCT AND LLM-THINKING MODELS

The rapid progress of LLMs has not only improved their instruction-following ability but also expanded their scope toward explicit reasoning. This shift has given rise to two major families: instruction-following models (**LLM-instruct**), which are trained to directly map user prompts into concise outputs, and reasoning-augmented models (**LLM-think**), which are trained to generate long chains of thought (CoT) reasoning (Wei et al., 2022) before arriving at final answers.

Although the two are built upon the same underlying architecture, recent studies show that they diverge significantly in both internal representations and emergent behaviors. For instance, LLM-think models retain richer contextual information across layers and exhibit higher token-level entropy (Skean et al., 2025; Wang et al., 2025). In terms of attention, LLM-instruct models generally display diverse specialization across heads, with each attending to different token subsets (Fu et al., 2024), whereas LLM-think models show considerable overlap in the key tokens attended by different heads (Yang et al., 2025b). Moreover, LLM-think models are highly sensitive to compression: pruning and distillation pipelines that perform well on language modeling often lead to substantially larger drops in accuracy on complex reasoning tasks (Zhang et al., 2025). These evidences suggest that *LLM-think models cannot be regarded as a straightforward extension of LLM-instruct.*

Despite these differences, nearly all existing pruning studies have been conducted exclusively on LLM-instruct (Kim et al., 2024; Men et al., 2024). *This is a critical omission*: pruning is arguably even more consequential for LLM-think, since their long reasoning traces impose high computational and memory costs (Chen et al., 2025), whereas LLM-instruct typically handles tasks with long inputs but shorter outputs (Zhou et al., 2023a). Yet it remains unclear whether pruning strategies validated on LLM-instruct can generalize to LLM-think, or whether new methods are needed. In this work, we take the **first** step toward addressing this gap. By leveraging settings where both models and their training data are fully accessible, we systematically examine how pruning interacts with instruction-following and reasoning behaviors alike. *Crucially, our study moves beyond simple benchmarking:* it delivers practical recommendations for the efficient deployment of both LLM-instruct and LLM-think, offering broader insights into how efficiency techniques must evolve alongside the changing landscape of LLMs.

### 2.2 FORMALIZING STRUCTURE PRUNING STRATEGIES

Model pruning in LLMs can be broadly categorized into two approaches: *unstructured pruning*, which removes individual weights based on their magnitudes or importance (Liao et al., 2023), and *structured pruning*, which discards entire groups such as neurons, heads, or layers (Cheng et al., 2024). Although unstructured pruning is conceptually simple, it rarely yields practical acceleration on modern GPUs (e.g., Nvidia GPUs typically require over 90% sparsity for speedup) , while LLMs usually collapse once sparsity exceeds 50% (Song et al., 2024). Therefore, we focus on structured pruning, which is both hardware-friendly and effective (Men et al., 2024). Structured pruning can be further categorized into *width pruning*, which reduces hidden dimensions by removing neurons or feature channels, and *depth pruning*, which removes redundant layers either statically (the same set of layers is pruned for all inputs) or dynamically (the pruned layers vary depending on the input). The architectures of each method are illustrated in Figure 1. To the best of our knowledge, a unified and systematic formulation of these strategies has not been explicitly articulated in the existing literature. We therefore introduce the following formal definitions, which serve as the foundation for our subsequent analysis.

Formally, let $\mathbf{H}_l \in \mathbb{R}^{N \times d}$ denote the hidden representations at layer $l$, where $N$ is the sequence length and $d$ is the hidden dimension.

- **Static width** pruning reduces the hidden dimension $d$ to $d' < d$ by removing less important neurons:
$$\mathbf{H}'_l = \mathbf{H}_l[:, \mathcal{I}_l], \quad |\mathcal{I}_l| = d', \tag{1}$$
where $\mathcal{I}_l$ indexes the retained neurons in layer $l$.

- **Static depth** pruning removes entire layers, keeping only a subset $\mathcal{L}' \subseteq \{1, \ldots, L\}$ of layers:
$$\mathbf{H}_{\mathcal{L}'} = \{\mathbf{H}_l \mid l \in \mathcal{L}'\}, \quad |\mathcal{L}'| = L' < L. \tag{2}$$

Static depth pruning computes a fixed importance score for each layer and permanently removes those identified as redundant. Typical criteria for scoring include:

– *Block Influence(BI)*(Men et al., 2024) measures the cosine similarity between the input and output representations of a layer. Layers with high similarity are considered to contribute little new information and are thus pruned.

– *Perplexity(PPL)*(Kim et al., 2024) quantifies the impact of individually removing a layer on validation perplexity. Layers that cause minimal degradation are considered less critical.

– *Taylor*(Kim et al., 2024) estimates the sensitivity of the loss to parameter removal through the first-order gradient–weight product, and the layer with low aggregated sensitivity are pruned.

• **Dynamic depth** pruning introduces a router module, denoted as $R(\cdot)$, which determines whether a block is executed or skipped for each input token (Raposo et al., 2024; Jiang et al., 2024; Zhao et al., 2025). Let $x$ denote the input to a block, which may correspond to a Transformer layer, an attention module, or a MLP. A binary gate $g$ governs the execution, and is defined as

$$g = R(x) \in \{0, 1\}. \tag{3}$$

where $g = 1$ indicates execution and $g = 0$ indicates skipping. Let $f(\cdot)$ denote the computation performed by the block; the output is then updated as:

$$x' = g \cdot f(x) + x. \tag{4}$$

### 2.3 PROBLEM SETTING

We formalize the pruning problem as follows. Let $M$ denote an LLM, which can be either an LLM-instruct or an LLM-think model. Our goal is to obtain a smaller, compressed model $M'$ through pruning. Although pruning and compression have been extensively studied, in the context of LLMs there is still no universally adopted metric for characterizing the degree of pruning. For clarity, we define the **compression ratio** as the ratio of the average number of model parameters used per token after pruning to that before pruning:

$$R(M, M') = 1 - \frac{|M'|}{|M|}, \tag{5}$$

where $|M|$ and $|M'|$ denote the average per-token parameters in the original and pruned models, respectively, and a higher $R$ indicates greater compression. We aim to maximize the performance of $M'$ on a set of unseen evaluation benchmarks $\mathcal{D}_{\text{eval}}$, where $\text{Perf}(M', \mathcal{D}_{\text{eval}})$ denotes a composite score that aggregates results across all benchmarks. The pruning problem can thus be formulated as a compression-constrained optimization:

$$\max_{M'} \ \text{Perf}(M', \mathcal{D}_{\text{eval}}) \quad \text{s.t.} \ \rho(M, M') \leq R_{\text{target}}, \tag{6}$$

where $R_{\text{target}} \in (0, 1)$ is the user-specified target compression ratio. In this work, we investigate this optimization problem under various pruning strategies, instantiating $\mathcal{D}_{\text{eval}}$ with benchmarks that test instruction-following for LLM-instruct and reasoning for LLM-think. Complementary to compression ratio, we further define **performance retention** as $\frac{\text{Perf}(M')}{\text{Perf}(M)}$, which quantifies how well the pruned model preserves the performance of the original dense model.

## 3 EXPERIMENTAL SETUP

### 3.1 MODEL

Our study focuses on two representative families of LLMs: **LLM-instruct** (instruction-following) and **LLM-think** (reasoning-oriented). For LLM-instruct, we adopt `Llama-3.1-Tulu-3-8B-SFT` (Lambert et al., 2024), an open-source model that releases both the weights and its instruction-tuning corpus together with detailed training configurations. For LLM-think, we instantiate a reasoning-oriented counterpart by fine-tuning Llama-3.1-8B-Instruct on the OpenThoughts dataset (Guha et al., 2025), yielding

`Llama-3.1-8B-Instruct-OpenThoughts`.[2] Both models share the same `Llama-3.1-8B` backbone, ensuring a controlled comparison of pruning effects. A distinctive feature of our setting is that the original training datasets for both families are fully accessible: they serve as calibration data during pruning and as recovery data for post-fine-tuning. Consequently, the pruned models are recovered under their native training distributions—rather than downstream task distributions—thus retaining their fundamental capabilities (instruction following for LLM-instruct and reasoning for LLM-think) instead of adapting to specific downstream tasks.

## 3.2 EVALUATION BENCHMARKS

To systematically evaluate the instruction-following capabilities of the LLM-instruct model and the reasoning capabilities of the LLM-think model, we use a diverse suite of **17** tasks that can be broadly divided into instruction-following and reasoning benchmarks.

- **Instruction Following Benchmarks.** To comprehensively assess LLM-instruct on instruction following, we include both classification and generation tasks. For *classification* tasks—which test whether the model can follow restricted answer options and correctly interpret the input— following (Touvron et al., 2023), we include BoolQ (Clark et al., 2019), PIQA (Bisk et al., 2020), HellaSwag (Zellers et al., 2019), WinoGrande (Sakaguchi et al., 2021), ARC-Easy / Challenge (ARC-E / C) (Clark et al., 2019), and OpenBookQA (OPQA) (Mihaylov et al., 2018). For *generation* tasks—designed to evaluate the ability to produce high-quality, coherent text while adhering to complex instructions—we adopt a similar setup to (Lambert et al., 2024), but exclude tasks that rely heavily on explicit CoT reasoning (evaluated separately under reasoning benchmarks). Specifically, we include IFEval (IFE) (Zhou et al., 2023b) to measure instruction-execution precision; TruthfulQA (TQA) (Lin et al., 2021) and PopQA (PQA) (Mallen et al., 2022) to assess factual accuracy and truthfulness; and HumanEval (HE) (Chen et al., 2021) together with HumanEval+ (HE+) (Liu et al., 2023) as constrained code-generation tasks, where strict adherence to problem specifications is critical.

- **Reasoning Benchmarks.** To rigorously evaluate the problem-solving abilities of the LLM-think model, we employ five challenging benchmarks spanning mathematical, coding, and scientific domains. Specifically, AIME 2024 (AIME) and MATH-500 (MATH) (Lightman et al., 2023) assess advanced mathematical reasoning and multi-step derivation; LiveCodeBench (LCB) (Jain et al., 2024) evaluates code generation, debugging, and comprehension in complex programming tasks; and GPQA-Diamond (GPQA) (Rein et al., 2024) together with JEEBench (JEE) (Arora et al., 2023) assess nuanced scientific reasoning and the application of domain-specific knowledge.

## 3.3 EVALUATED PRUNING METHODS

To systematically evaluate pruning in both LLM-instruct and LLM-think models, we consider representative methods from three main categories of structured pruning, as described in Section 2.2.

- Static width pruning methods reduce the model width by removing redundant parameters. *LLM-Pruner* is a gradient-based method that prunes unimportant coupled structures (Ma et al., 2023), while *SliceGPT* removes low-variance components from weight matrices through principal component analysis (PCA) (Ashkboos et al., 2024).

- Static depth pruning methods reduce the depth of the model by removing layers. *ShortGPT* (Men et al., 2024) prunes entire layers using BI, which is based on input-output cosine similarity. *Shortened-llama-PPL* and *Shortened-llama-Taylor* (Kim et al., 2024) evaluate and remove layers based on a combination of PPL and Taylor expansion.

- Dynamic depth pruning adaptively skips layers, attention blocks, or MLP modules for each input. *MOD* (Raposo et al., 2024) dynamically selects a subset of tokens for computation in each layer using a Top-k routing mechanism. *D-LLM* (Jiang et al., 2024) employs a router module to adaptively skip each transformer layer. *SkipGPT* (Zhao et al., 2025) is a dynamic framework combining global token-aware routing with decoupled pruning for MLP and self-attention layers.

---

[2]Llama-3.1-8B-Instruct-OpenThoughts is obtained by fine-tuning Llama-3.1-8B-Instruct on OpenThoughts (Guha et al., 2025) using Llama-Factory (Zheng et al., 2024). Training was performed on $8\times$ H20 (96 GB) GPUs for 3 epochs ($\approx 488$ GPU hours). See Table 4 for details. To our knowledge, it is the first reasoning model trained on a fully open corpus, with both the model and its training data publicly released.

Table 1: Performance on classification tasks under different pruning ratio. The dense baseline is Llama 3.1-Tulu-3-8B-SFT(LLM-insturct). For each pruning ratio, the best result is marked in **bold**, and the second-best is underlined. Color coding indicates pruning strategy: Static Depth Pruning, Static Width Pruning, and Dynamic Depth Pruning.

| Ratio | Method | BoolQ Acc | OBQA AccNorm | PIQA Acc | WinoGrande Acc | HeSw AccNorm | ARC-E AccNorm | ARC-C AccNorm | Avg. Acc.↑ |
|---|---|---|---|---|---|---|---|---|---|
| 0.00% | Dense | 82.26 | 46.80 | 80.84 | 77.74 | 82.97 | 87.20 | 61.43 | 74.18 |
| 20.0% | ShortGPT | 68.34 | 39.60 | 74.59 | 75.84 | 75.16 | 79.41 | 51.53 | 66.35 |
| | Shortened-PPL | 68.16 | 43.20 | 78.12 | 64.56 | 71.01 | 78.28 | 48.12 | 64.49 |
| | Shortened-Taylor | 74.83 | 42.80 | 76.49 | 77.03 | 77.62 | 80.05 | 53.92 | 68.96 |
| | LLM-Pruner | 71.71 | 38.40 | 75.40 | 62.98 | 68.86 | 74.11 | 43.77 | 62.18 |
| | SliceGPT | 80.73 | 34.80 | 72.63 | 69.77 | 68.45 | 71.96 | 44.79 | 63.59 |
| | MOD | 69.78 | 36.00 | 72.85 | 66.14 | 73.43 | 74.62 | 47.95 | 62.40 |
| | D-LLM | 64.64 | 27.20 | 58.65 | 56.19 | 60.66 | 64.52 | 37.71 | 52.80 |
| | SkipGPT | 80.39 | 47.20 | 77.80 | 74.11 | 78.62 | 85.56 | 60.40 | 72.30 |
| 40.0% | ShortGPT | 67.82 | 29.40 | 67.62 | 68.19 | 60.00 | 60.85 | 39.07 | 56.99 |
| | Shortened-PPL | 45.65 | 33.80 | 71.81 | 53.98 | 56.50 | 66.87 | 35.32 | 50.85 |
| | Shortened-Taylor | 73.60 | 30.80 | 69.04 | 70.40 | 63.31 | 65.15 | 40.52 | 58.97 |
| | LLM-Pruner | 63.24 | 30.80 | 66.53 | 55.01 | 48.05 | 56.31 | 30.80 | 50.11 |
| | SliceGPT | 74.77 | 29.80 | 63.65 | 61.01 | 51.02 | 55.93 | 33.87 | 52.29 |
| | MOD | 64.18 | 32.20 | 69.85 | 62.27 | 65.67 | 68.98 | 42.49 | 57.66 |
| | D-LLM | 58.13 | 26.60 | 52.72 | 54.14 | 41.58 | 46.96 | 28.92 | 44.86 |
| | SkipGPT | 81.74 | 41.20 | 77.31 | 75.29 | 82.01 | 86.44 | 60.58 | 72.94 |
| 60.0% | ShortGPT | 60.27 | 26.60 | 57.23 | 52.40 | 35.56 | 35.56 | 23.12 | 41.53 |
| | Shortened-PPL | 60.42 | 28.00 | 63.11 | 51.53 | 32.81 | 49.24 | 26.36 | 44.21 |
| | Shortened-Taylor | 55.41 | 27.80 | 60.33 | 55.80 | 38.47 | 41.75 | 24.74 | 43.19 |
| | LLM-Pruner | 53.51 | 26.20 | 60.88 | 51.38 | 32.41 | 41.49 | 20.90 | 40.11 |
| | SliceGPT | 63.12 | 26.40 | 57.72 | 52.56 | 35.44 | 39.23 | 23.98 | 42.92 |
| | MOD | 59.69 | 27.80 | 55.60 | 54.06 | 47.17 | 48.48 | 31.14 | 46.56 |
| | D-LLM | 57.06 | 24.80 | 52.06 | 50.43 | 31.98 | 37.20 | 23.97 | 39.64 |
| | SkipGPT | 83.21 | 39.60 | 77.14 | 73.79 | 81.64 | 86.32 | 60.66 | 71.77 |

# 4 EXPERIMENTS

In this section, we present a comprehensive study of three pruning strategies applied to the LLM-instruct and LLM-think models. Specifically, both models and datasets are fully accessible, we used Tulu-Mixture-SFT as the calibration and post-fine-tuning recovery dataset for Llama-3.1-Tulu-3-8B (LLM-instruct), and similarly employed OpenThoughts for Llama-3.1-8B-instruct-OpenThoughts (LLM-think). For evaluation, *each model was tested on tasks aligned with its respective capabilities: LLM-instruct on instruction-following benchmarks, and LLM-think on reasoning benchmarks.*

This setup allows us to answer several key questions: 1) Among dynamic depth, static depth, and static width pruning strategies, which is most effective, and is this ranking consistent across tasks? 2) Can pruning strategies developed for LLM-instruct be directly transferred to LLM-think? 3) Which model exhibits greater sensitivity to pruning within its domain of expertise? 4) Does leveraging the native training distribution enable more effective recovery of a model's performance after pruning?

## 4.1 PRUNING STRATEGIES: PERFORMANCE ACROSS DIVERSE TASKS

In this subsection, we examine how the three pruning strategies interact with the three task types—classification, generation, and reasoning—and investigate whether pruning can fully restore each model's capabilities within its native training distribution.

**Static depth vs. width pruning.** As shown in Figure 2, both static pruning strategies achieve similar performance on classification tasks. However, as shown in Table 3, for the generation and reasoning tasks, a clear trend emerges: as the pruning ratio increases, static width pruning exhibits a notably slower degradation in performance compared to static depth pruning. For instance, as reported in Table 1, with the pruning ratio 20%, both static pruning strategies perform similarly on generation tasks (48.72 vs. 48.02). When the pruning ratio increases to 40%, the performance of static depth pruning drops by an average of 55.64%, while static width pruning degrades by 41.51%. Furthermore, while static depth pruning achieves better performance than static width pruning at 20% pruning (20.23 vs. 17.78), both methods experience severe degradation at 40%, with static depth pruning dropping by 88.40% compared to 77.69% for static width pruning. These results

Table 2: Performance on generation (w/o CoT) and reasoning tasks under different pruning ratio. For each pruning ratio, the best result is in **bold**, and the second-best is underlined.

| Ratio | Method | LLM-instruct: Generation (w/o CoT) | | | | | Avg.↑ | LLM-think: Reasoning Tasks | | | | | Avg.↑ |
|---|---|---|---|---|---|---|---|---|---|---|---|---|---|
| | | IFE Pr. | TQA mc2 | PQA | HE p@10 | HE+ p@10 | | MATH Acc | AIME Acc | LCB Acc | GPQA Acc | JEE Acc | |
| 0.00% | Dense | 74.12 | 46.78 | 29.44 | 84.22 | 77.49 | 62.41 | 71.80 | 20.00 | 10.03 | 42.42 | 32.33 | 35.72 |
| 20.0% | ShortGPT | 60.81 | 45.24 | 13.16 | 66.82 | 60.88 | 49.38 | 52.00 | 3.33 | **3.25** | 23.74 | **19.37** | 20.34 |
| | Shortened-PPL | 48.42 | 36.00 | **21.55** | 42.77 | 40.33 | 37.81 | 53.00 | 0.00 | 1.63 | **29.80** | 17.86 | **20.46** |
| | Shortened-Taylor | 65.61 | 45.47 | 15.31 | 80.63 | 73.36 | 56.88 | 53.40 | 3.33 | 0.00 | 23.74 | 19.08 | 19.91 |
| | LLM-Pruner | 52.49 | 42.62 | 15.16 | 57.29 | 53.34 | 44.58 | 52.30 | 3.33 | 1.85 | 22.50 | 5.35 | 17.07 |
| | SliceGPT | 63.58 | 44.74 | 10.77 | 74.71 | 70.47 | 52.85 | **60.00** | 3.33 | 0.00 | 22.50 | 6.69 | 18.50 |
| | MOD | **68.20** | 44.74 | 14.75 | 82.18 | **78.40** | 57.65 | 0.00 | 0.00 | 0.00 | 2.25 | 0.00 | 0.45 |
| | D-LLM | 29.75 | 42.63 | 10.94 | 26.68 | 20.47 | 26.49 | 1.00 | 0.00 | 0.00 | 15.00 | 4.17 | 4.03 |
| | SkipGPT | 67.09 | 47.03 | 17.10 | 83.65 | 76.12 | 58.20 | 0.00 | 0.00 | 0.00 | 13.63 | 0.00 | 2.73 |
| 40.0% | ShortGPT | 31.60 | 46.89 | 9.44 | 13.71 | 11.88 | 22.70 | 3.40 | 0.00 | 0.00 | 15.15 | **4.27** | 4.56 |
| | Shortened-PPL | 27.91 | 37.34 | 13.49 | 18.92 | 15.15 | 22.36 | 0.40 | 0.00 | 0.00 | 13.64 | 2.52 | 3.31 |
| | Shortened-Taylor | 50.09 | 43.69 | 10.92 | 46.18 | 41.11 | 38.00 | 3.80 | 0.00 | 0.00 | 16.16 | 2.86 | 4.56 |
| | LLM-Pruner | 36.41 | 42.98 | 9.60 | 23.90 | 22.19 | 27.42 | 0.00 | 3.33 | 0.00 | 20.00 | 0.00 | 4.67 |
| | SliceGPT | 57.30 | 48.12 | 6.83 | 60.59 | 55.10 | 45.59 | 29.00 | 0.00 | 0.00 | 25.00 | 2.37 | **11.27** |
| | MOD | 48.79 | 42.26 | 14.08 | 72.23 | 67.77 | 49.03 | 0.00 | 0.00 | 0.00 | 0.00 | 0.00 | 0.00 |
| | D-LLM | 20.14 | 44.39 | 7.63 | 11.63 | 8.67 | 18.89 | 3.00 | 0.00 | 0.00 | 17.50 | 3.78 | 4.86 |
| | SkipGPT | **70.61** | 50.04 | 23.88 | 83.09 | 75.67 | 60.66 | 0.00 | 0.00 | 0.00 | 10.33 | 0.00 | 2.07 |
| 60.0% | ShortGPT | 10.90 | 47.10 | 5.69 | 0.00 | 0.30 | 12.40 | 0.00 | 0.00 | 0.00 | 1.51 | 0.00 | 0.30 |
| | Shortened-PPL | 18.11 | 41.99 | 5.40 | 7.37 | 6.17 | 15.01 | 0.00 | 0.00 | 0.00 | 12.46 | 0.00 | 2.49 |
| | Shortened-Taylor | 17.56 | 43.74 | 6.46 | 6.99 | 5.13 | 15.18 | 0.00 | 0.00 | 0.00 | **16.66** | 0.00 | 3.33 |
| | LLM-Pruner | 20.14 | 47.19 | 2.29 | 6.58 | 6.39 | 16.92 | 0.00 | 0.00 | 0.00 | 13.75 | 0.00 | 2.75 |
| | SliceGPT | 35.12 | 47.54 | 6.61 | 29.00 | 25.59 | 28.77 | 3.00 | 0.00 | 0.00 | 2.00 | 0.67 | 1.13 |
| | MOD | 10.35 | 44.16 | 3.73 | 55.93 | 48.39 | 32.91 | 0.00 | 0.00 | 0.00 | 0.00 | 0.00 | 0.00 |
| | D-LLM | 11.46 | 47.22 | 3.74 | 2.31 | 2.10 | 13.77 | 3.00 | 0.00 | 0.00 | 11.25 | 5.38 | **3.93** |
| | SkipGPT | **69.68** | 45.19 | **22.97** | **83.15** | **77.25** | 59.65 | 0.00 | 0.00 | 0.00 | 2.83 | 0.00 | 0.57 |

Table 3: Performance decline of pruning methods on classification, generation, and reasoning tasks. The row Dense shows the unpruned model performance. Rows marked **AD (avg)** shows the average relative performance drop of methods within the same pruning strategy at each sparsity level.

| | Classification | | | Generation (w/o CoT) | | | Reasoning | | |
|---|---|---|---|---|---|---|---|---|---|
| Dense | 74.18 | | | 62.41 | | | 35.72 | | |
| Sparsity | 0.2 | 0.4 | 0.6 | 0.2 | 0.4 | 0.6 | 0.2 | 0.4 | 0.6 |
| ShortGPT | $66.35_{\downarrow 10.56}$ | $56.99_{\downarrow 23.17}$ | $41.53_{\downarrow 44.01}$ | $49.38_{\downarrow 20.88}$ | $22.70_{\downarrow 63.63}$ | $12.40_{\downarrow 80.13}$ | $20.34_{\downarrow 43.06}$ | $4.56_{\downarrow 87.23}$ | $0.30_{\downarrow 99.16}$ |
| Shortened-PPL | $64.49_{\downarrow 13.06}$ | $50.85_{\downarrow 31.45}$ | $44.21_{\downarrow 40.40}$ | $37.81_{\downarrow 39.42}$ | $22.36_{\downarrow 64.17}$ | $15.01_{\downarrow 75.95}$ | $20.46_{\downarrow 42.72}$ | $3.31_{\downarrow 90.73}$ | $2.49_{\downarrow 93.03}$ |
| Shortened-Taylor | $68.96_{\downarrow 7.04}$ | $58.97_{\downarrow 20.50}$ | $43.19_{\downarrow 41.78}$ | $56.88_{\downarrow 8.86}$ | $38.00_{\downarrow 39.11}$ | $15.18_{\downarrow 75.68}$ | $19.91_{\downarrow 44.26}$ | $4.56_{\downarrow 87.23}$ | $3.33_{\downarrow 90.68}$ |
| **AD (avg)** | 10.22% | 25.04% | 42.06% | 23.05% | 55.64% | 77.25% | 43.35% | 88.40% | 94.29% |
| LLM-Pruner | $62.18_{\downarrow 16.18}$ | $50.11_{\downarrow 32.45}$ | $40.11_{\downarrow 45.93}$ | $44.58_{\downarrow 28.57}$ | $27.42_{\downarrow 56.06}$ | $16.92_{\downarrow 72.89}$ | $17.07_{\downarrow 52.21}$ | $4.67_{\downarrow 86.93}$ | $2.75_{\downarrow 92.30}$ |
| SliceGPT | $63.59_{\downarrow 14.28}$ | $52.29_{\downarrow 29.51}$ | $42.92_{\downarrow 42.14}$ | $52.85_{\downarrow 15.32}$ | $45.59_{\downarrow 26.95}$ | $28.77_{\downarrow 53.90}$ | $18.50_{\downarrow 48.21}$ | $11.27_{\downarrow 68.45}$ | $1.13_{\downarrow 96.84}$ |
| **AD (avg)** | 15.23% | 30.98% | 44.04% | 21.95% | 41.51% | 63.40% | 50.21% | 77.69% | 94.57% |
| SKIPGPT | $72.30_{\downarrow 2.53}$ | $72.94_{\downarrow 1.67}$ | $71.77_{\downarrow 3.25}$ | $58.20_{\downarrow 6.75}$ | $60.66_{\downarrow 2.80}$ | $59.65_{\downarrow 4.42}$ | $2.73_{\downarrow 92.36}$ | $2.07_{\downarrow 94.20}$ | $0.57_{\downarrow 98.40}$ |
| mod | $62.40_{\downarrow 15.88}$ | $57.66_{\downarrow 22.27}$ | $46.56_{\downarrow 37.23}$ | $57.65_{\downarrow 7.63}$ | $49.03_{\downarrow 21.44}$ | $32.91_{\downarrow 47.27}$ | $0.45_{\downarrow 98.74}$ | $0.00_{\downarrow 100.00}$ | $0.00_{\downarrow 100.00}$ |
| dllm | $52.80_{\downarrow 28.82}$ | $44.86_{\downarrow 39.53}$ | $39.64_{\downarrow 46.56}$ | $26.49_{\downarrow 57.55}$ | $18.89_{\downarrow 69.73}$ | $13.77_{\downarrow 77.94}$ | $4.03_{\downarrow 88.72}$ | $4.86_{\downarrow 86.39}$ | $3.93_{\downarrow 89.00}$ |
| **AD (avg)** | 15.74% | 21.16% | 29.01% | 23.98% | 31.32% | 43.21% | 93.27% | 93.53% | 95.80% |

indicate that, in both generation and reasoning tasks, pruning along the depth dimension degrades performance more severely than pruning along the width dimension.

**Dynamic vs. Static pruning.** Results on classification and generation tasks reveal that dynamic depth pruning achieves consistent gains over both static pruning at all pruning ratios. At a 60% pruning ratio, the best dynamic method (SkipGPT) retains over 95% of the performance of instruction following capabilities, highlighting its strong robustness, while the best static method (SliceGPT) drops below 50%. Overall, *these results indicate that for classification and generation tasks, dynamic depth achieves the highest performance, followed by static width and then static depth.*

However, at a pruning ratio of only 20%, dynamic depth pruning retains merely 6.73% of the original reasoning performance, making it almost entirely ineffective. In contrast, static methods retain more than 53.46% of their performance at the same pruning ratio. Among them, static width pruning demonstrates the greatest robustness, maintaining 31.55% performance and some reasoning ability

even at a 40% ratio. *Taken together, these results show that for reasoning tasks, static width pruning is the most effective, while static depth pruning is weaker and dynamic depth pruning lags far behind.*

These experiments demonstrate that *no single pruning strategy is universally optimal*. The performance of different strategies varies across tasks: while dynamic depth pruning achieves strong results on classification and generation, it fails to transfer effectively to reasoning tasks.

> **Conclusion**
>
> The optimal pruning strategy depends on the task type. Dynamic depth pruning is most effective for classification and generation, while static width pruning shows the greatest robustness in reasoning. Static depth pruning consistently lags behind.

### 4.2 SENSITIVITY OF LLM-INSTRUCT AND LLM-THINK MODELS TO PRUNING RATIOS

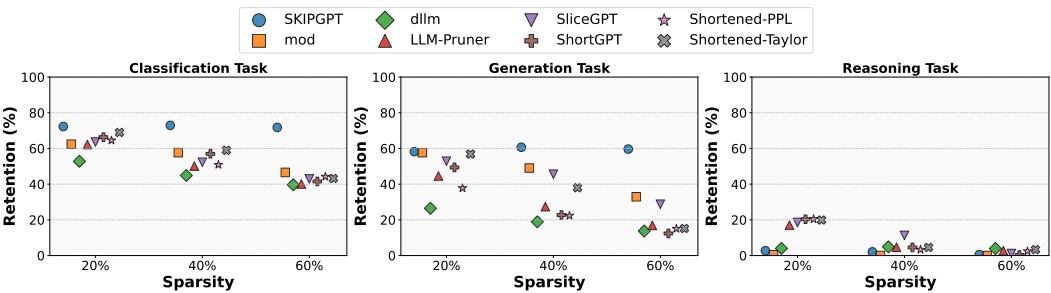

Figure 2: Performance of different pruning methods under varying pruning ratios on classification, generation, and reasoning tasks.

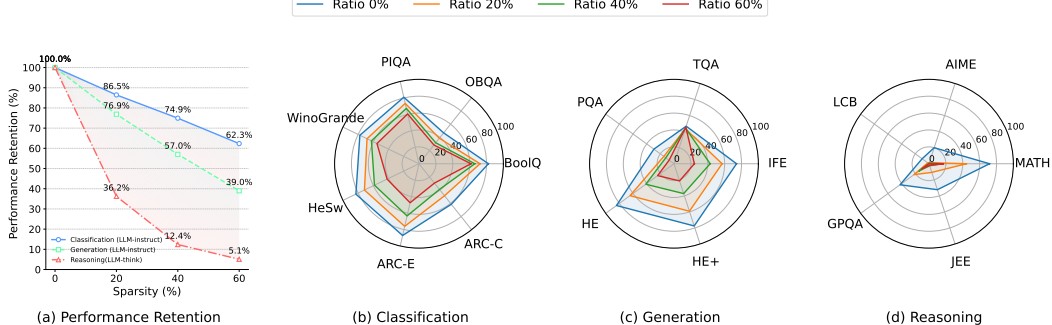

Figure 3: The average impact of different pruning ratios on model performance: (a) Performance Retention (b) generation, (c) classification, and (d) reasoning. For each pruning ratio, the performance score represents the average across all pruning methods at that ratio.

The previous section revealed a clear task dependency in pruning strategy performance, raising a key question: does this discrepancy stem from a mismatch between method and task, or from fundamental model differences? To investigate, we examine sensitivity within each model's domain of expertise. Specifically, we measure performance degradation of LLM-instruct on instruction-following benchmarks and LLM-think on reasoning benchmarks under identical pruning strategies and ratios.

Our experiments first uncover a divergence within the LLM-instruct model. As shown in Figure 2, both classification and generation tasks experience an linear decrease in performance, with the slope substantially steeper for generation. With a 60% pruning ratio, classification retains approximately 62.3% of its original performance, while generation drops to 39.8%. We attribute this difference to task-specific structural dependencies: classification tasks rely on redundant global semantic representations, whereas generation tasks are highly sensitive to disruptions in local sequential dependencies, where even small perturbations can propagate and significantly affect output quality.

Furthermore, our experiments reveal fundamental differences between the two model paradigms under pruning. As shown in Figure 2, their behaviors under pruning are strikingly different: LLM-instruct demonstrates high robustness, with near-linear, smooth, and predictable performance degradation as sparsity increases. In contrast, as shown in Figure 3, the outer blue polygon with high scores on the MATH axis indicates strong reasoning ability. Once pruning is applied, however, LLM-think shows extreme sensitivity, with reasoning performance collapsing in a cliff-like manner. Specifically, at 20% pruning, the performance retention rate of LLM-instruct models is reduced to below 60%, and at 40% sparsity, their complex reasoning ability is almost entirely lost. Overall, *LLM-think is substantially more sensitive to pruning ratios than LLM-instruct, indicating that models optimized for reasoning are far less stable than instruction following models.*

> **Conclusion**
>
> Pruning affects LLM-instruct and LLM-think models in fundamentally different ways. LLM-instruct models are relatively robust to pruning, whereas LLM-think models are highly sensitive: even light pruning can cause in logical errors and catastrophic failures.

### 4.3 The Effective of Calibration and Post-Fine-tuning Datasets

We conduct a series of experiments to examine whether using the model's native training data for pruning calibration, combined with post-fine-tuning, can effectively restore its general capabilities. Since our focus in this section is on the effect of calibration and post-fine-tuning datasets rather than a comparison of pruning methods, we select ShortGPT—the best-performing static depth pruning strategy in our previous experiments—as a representative method. We then apply it to the LLM-instruct model at a 20% pruning ratio, using four calibration datasets: Tulu-mixture-SFT (the native training dataset of Llama-3.1-Tulu-3-8B-SFT), C4 (Raffel et al., 2020), BookCorpus (Zhu et al., 2015), and OpenThoughts. As shown in Table 5, these four calibrations yielded only two distinct pruned models, indicating that some calibration data produce identical layer selection results. These observations imply that *without recovery fine-tuning, simply changing the calibration dataset has a negligible effect on the performance of the pruned model.*

Having established that calibration datasets have minimal effect, we turn to the choice of recovery training datasets in the post-fine-tuning stage. We employ Tulu-Mixture-SFT and Alpaca, while the pruned model is obtained from calibration on Tulu-Mixture-SFT (BookCorpus) and C4 (OpenThoughts). As shown in Table 6, post-fine-tuning with the original training dataset Tulu-Mixture-SFT achieves the best performance, demonstrating that *alignment with the original data distribution is essential for effectively recovering the performance of the pruned model.*

> **Conclusion**
>
> Effective recovery of pruned models depends mainly on post-fine-tuning with datasets aligned to the model's original training distribution, whereas calibration data choice has little impact.

## 5 Conclusion

We present the first systematic study of pruning across instruction-following (**LLM-instruct**) and reasoning-augmented (**LLM-think**) models. Leveraging open training corpora, we build an experimental framework for pruning and recovery within the original data distribution, and release `Llama-3.1-8B-Instruct-OpenThoughts`, the first reasoning model trained on a fully open corpus. Our results show that pruning effectiveness is task- and model-dependent. Dynamic depth pruning is most effective for classification and generation, while static width pruning is most robust for reasoning, with static depth consistently lagging. Strategies designed for LLM-instruct do not transfer to LLM-think, which proves far more sensitive to pruning. Effective recovery relies mainly on post-fine-tuning with data aligned to the original training distribution, whereas calibration data matter little. Overall, pruning interacts deeply with model family, task type, and data distribution, offering guidance for compressing reasoning-augmented LLMs.

REPRODUCIBILITY STATEMENT

We have made every effort to ensure the reproducibility of our results. All experiments were conducted using the official repositories introduced in these papers. The training configurations, including hyperparameters, are detailed in Table 4. We believe these measures will enable other researchers to reproduce our results and build upon our work.

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

# A  THE USE OF LARGE LANGUAGE MODELS

Large Language Models (LLMs) were used to support the writing and editing of this manuscript. Specifically, we employed an LLM to refine the language, improve readability, and enhance clarity in selected sections. The model was used for tasks such as sentence rephrasing, grammar correction, and improving the overall flow of the text.

The LLM was not involved in generating the study's ideas, designing the research methodology, conducting experiments, analyzing data, or interpreting results. All scientific concepts, methods, and analyses presented in this work were independently conceived and carried out by the authors.

The authors take full responsibility for the content of the manuscript, including portions refined with the assistance of the LLM. Its use followed ethical standards and did not contribute to plagiarism or scientific misconduct.

# B  RELATED WORK

The multi-layer Transformer architecture of LLMs inherently exhibits substantial parameter redundancy. Model pruning serves as a key technique to address this issue by eliminating non-essential components. In practice, however, unstructured pruning induces sparsity that is challenging to exploit on modern hardware, hardware-friendly structured pruning has emerged as the predominant approach. Structured pruning removes entire components such as channels, attention heads, or layers, and can be broadly categorized into static and dynamic methods.

## B.1  STATIC PRUNING

Static pruning aims to permanently remove parameters of a pretrained model to create a smaller dense model that is efficient across inputs. This approach can be divided into width and depth pruning based on the dimension of removal.

**Width pruning** focuses on reducing the width of the network by removing components within each layer, such as attention heads, MLP neurons, or coupled structures. A central challenge is to define the importance criteria for these components. Some methods leverage gradient information to identify and eliminate unimportant coupled structures (Ma et al., 2023). Other comprehensive approaches perform end-to-end pruning across layers, attention heads, and hidden dimensions simultaneously (Xia et al., 2023). More recent works have explored training-free criteria; for instance, Wanda prunes channels based on the product of weights and input activations without requiring retraining (Sun et al., 2023), while others use fluctuation-based importance metrics (An et al., 2024). Beyond importance-based pruning, another direction exploits computational invariance in Transformers, using PCA to remove minor components and densify weight matrices (Ashkboos et al., 2024).

**Depth pruning** offers a more direct compression strategy by removing entire Transformer layers, thereby reducing model depth. The main challenge is to assess layer importance accurately to avoid severe performance degradation. Researchers have proposed various metrics to this end, such as measuring the cosine similarity between a block's input and output to quantify its influence (Men et al., 2024), or leveraging the high similarity between adjacent blocks to remove redundant ones (Song et al., 2024). Others combine perplexity (PPL) with Taylor expansion methods to evaluate and remove multiple layers at once (Kim et al., 2024). Furthermore, some strategies propose joint pruning of both attention and MLP modules within layers to achieve a better trade-off between compression and performance (He et al., 2024).

## B.2  DYNAMIC PRUNING

In contrast to static methods, dynamic pruning customizes the computational path for each input at inference time, reducing computation by executing only essential components.

A widely studied approach is *early exit* (Schuster et al., 2022; Varshney et al., 2023; Del Corro et al., 2023; Din et al., 2023; Chen et al., 2024; Fan et al., 2024). By adding intermediate classifiers at various depths of the model, early exit allows "simple" inputs to terminate inference prematurely, thus

bypassing the remaining layers. While effective for acceleration, this can potentially compromise the model's capacity for deep semantic reasoning.

Another paradigm is *layer skipping*, where router modules are employed to dynamically decide whether a layer should be executed or bypassed. For example, some methods propose dynamic computation allocation that uses a top-k routing mechanism to select which tokens are processed by each layer's self-attention and MLP modules (Raposo et al., 2024). D-LLM designs a dynamic decision module at each layer to adaptively execute network units and introduces an efficient eviction strategy to address the resulting KV cache challenges (Jiang et al., 2024). Similarly, SkipGPT proposes a framework that combines global token-aware routing with decoupled pruning strategies for MLP and self-attention layers to achieve fine-grained resource allocation (Zhao et al., 2025). Together, these methods offer a flexible way to balance inference efficiency and model performance on a per-input basis.

## C  EXPERIMENTS DETAILS

For LLM-instruct model, all experiments are conducted on a single A800 GPU without using Deep-Speed. We adopt Llama 3.1-Tulu-3-8B-SFT, obtained through supervised fine-tuning (SFT) on Llama 3.1-8B (Lambert et al., 2024), as our dense baseline model. The SFT training corpus is adopted both as the calibration set for model pruning and as the training set for subsequent LoRA fine-tuning. Specifically, we use a batch size of 16 and train all baselines for 10,000 steps. For each baseline, we conduct a grid search over learning rates to select the optimal value. We use a cosine decay learning rate schedule and set the warmup ratio to 0.1. The maximum token length is set to 4,096.

For the LLM-think model, no reasoning-oriented model exists with fully open-source training data. To address this gap, we fine-tuned Llama-3.1-8B-Instruct on the OpenThoughts dataset, resulting in a new model, Llama-3.1-3-8B-Instruct-OpenThoughts. The detailed hyperparameters are summarized in Table 4. For all baseline pruning methods, experiments are conducted on a single A800 GPU with DeepSpeed Stage-2 offloading and gradient checkpointing enabled. Following the setup in LLM-instruct experiments, the OpenThoughts dataset is used both as the calibration set for model pruning and as the training set for subsequent LoRA fine-tuning. Specifically, we adopt a batch size of 16 and train all baselines for 3,000 steps, applying early stopping if the training converges before reaching this limit. For each baseline, we conduct a grid search over learning rates to select the optimal value. We employ a cosine decay learning rate schedule and set the warmup ratio to 0.1. The maximum token length is set to 16,384.

Table 4: Full fine-tuning hyperparameters for training **LLaMA-3.1-8B-Instruct** on **OpenThoughts** to obtain the **LLM-think** model.

| Hyperparameter | Value |
| --- | --- |
| Max token length | 16,384 |
| Per-device train batch size | 1 |
| Per-device eval batch size | 8 |
| Gradient accumulation steps | 3 |
| Learning rate | $1 \times 10^{-5}$ |
| Number of training epochs | 3 |
| LR scheduler type | Cosine |
| Warmup ratio | 0.1 |
| Seed | 42 |
| Optimizer | AdamW (torch) |
| Weight decay | 0 |
| Adam $\beta_1$ | 0.9 |
| Adam $\beta_2$ | 0.999 |
| Adam $\epsilon$ | $1 \times 10^{-8}$ |
| Max gradient norm | 1.0 |
| bf16 precision | True |
| fp16 precision | False |

# D THE GENERATED TOKEN NUMBER AND PERFORMANCE IN HUMANEVAL AND HUMANEVALPLUS TASKS.

As the pruning ratio increases, we observe that the evaluation on HumanEval and HumanEval+ becomes more time-consuming. Figure 4 reports a comparative analysis of different pruning methods, where task performance (Score) is plotted against inference efficiency (Generated Tokens). An ideal pruning strategy should lie in the top-left region of the plot, reflecting high accuracy with low computational cost. The results highlight the effectiveness of SKIPGPT (green circles) and mod (purple circles), which achieve a favorable balance between performance and efficiency.

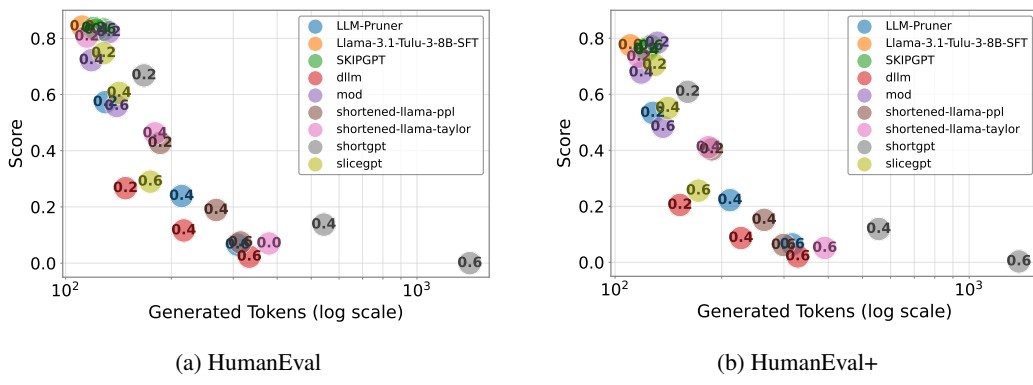

(a) HumanEval          (b) HumanEval+

Figure 4: Performance vs. Generated Tokens on (a) HumanEval and (b) HumanEval+.

# E CALIBRATION AND FINE-TUNING DATASETS OF BASELINES

Table 5: Different calibration datasets are used to guide the pruning of ShortGPT, and the pruned models (w/o LoRA) are subsequently evaluated.

| Calibrations / Tasks | BoolQ | OPQA | PIQA | Winogrande | Avg.↑ |
|---|---|---|---|---|---|
| Dense | 82.26 | 46.8 | 80.84 | 77.74 | 71.91 |
| Tulu | 69.33 | 39.4 | 70.18 | 72.30 | 62.80 |
| Bookcorpus | 69.33 | 39.4 | 70.18 | 72.30 | 62.80 |
| OpenThoughts | 76.76 | 38.2 | 72.09 | 73.09 | 65.03 |
| C4 | 76.76 | 38.2 | 72.09 | 73.09 | 65.03 |

Table 6: The post-fine-tuning performance of pruned models using Tulu-Mixture-SFT and Alpaca as recovery training datasets.

| with lora | Calibrations | Fine-tuning | PIQA | Winogrande | ARC-C | IFE | HE+ | Avg.↑ |
|---|---|---|---|---|---|---|---|---|
| Dense | - | - | 80.84 | 77.74 | 87.20 | 74.12 | 84.22 | 80.82 |
| ShortGPT | Tulu/BookCorpus | Tulu | 74.59 | 75.84 | 79.41 | 60.81 | 60.88 | 70.30 |
| ShortGPT | Tulu/BookCorpus | Alpaca | 75.68 | 75.61 | 81.36 | 52.86 | 60.27 | 69.15 |
| ShortGPT | C4/OpenThoughts | Alpaca | 76.39 | 74.66 | 81.69 | 53.60 | 58.61 | 68.99 |

In this section, we summarize the original calibrations and fine-tuning datasets adopted by the baseline models:

- **ShortGPT** employs PG19 (Rae et al., 2019) for computing BI scores and Samsum (Gliwa et al., 2019) for fine-tuning.

- **Shortened-llama-PPL** and **Shortened-llama-Taylor** utilize BookCorpus (Zhu et al., 2015) to estimate block scores (based on PPL or Taylor expansion) and Alpaca (Taori et al., 2023) for LoRA-based fine-tuning.

- **LLM-pruner** leverages BookCorpus (Zhu et al., 2015) to capture coupled structures and Alpaca (Taori et al., 2023) for LoRA fine-tuning.

- **SliceGPT** adopts either WikiText-2 (Merity et al., 2016) or Alpaca (Taori et al., 2023) for both calibration and fine-tuning.
- **SKIPGPT** uses RedPajama (Weber et al., 2024) for router tuning and applies a two-stage LoRA training procedure.

