# OpenReview forum: "From LLMs to LRMs: Rethinking Pruning for Reasoning-Centric Models"
_ICLR.cc/2026/Conference — ICLR 2026 Conference Withdrawn Submission_

### Official Review · Reviewer_VMn3 · 2025-10-30

**Soundness:** 1
**Presentation:** 2
**Contribution:** 1
**Rating:** 2
**Confidence:** 5

**Summary:**

This paper empirically study the effectiveness of model pruning on reasoning models. The authors apply existing structured pruning methods to a thinking model trained by themselves and report that the model performance collapses across most pruning ratios and benchmarks. While the topic is potentially interesting, the experiments are not rigorous, the methodology has serious conceptual flaws, and the presented results are unconvincing.

**Strengths:**

The paper explored the topic of pruning in thinking LLMs, which is less explored before.

**Weaknesses:**

- Severe Logical Flaw in Experimental Design. The paper’s core experiment misunderstands the fundamental goal of model pruning. The purpose of pruning is to obtain a smaller model that performs **better than or at least comparably to a model of equivalent size trained from scratch**. Even a small reasoning model (e.g., 1.5B parameters) [7] should retain non-zero performance instead of 0.0% across all benchmarks. If pruning leads to zero performance, **that strongly suggests implementation errors, misconfigured evaluation, or a fundamentally flawed experimental setup** (I will tell the authors the correct scheme to perform pruning later in weakness 3), not an inherent limitation of thinking models. The authors’ claim that pruning “destroys” thinking ability is therefore unreliable.
  - For example, if the authors prune an 8B model to 6.4B (20% sparsity), they must compare it to a **6.4B model trained from scratch** (e.g., a LLaMA-3.1 configuration adjusted to 6.4B parameters) fine-tuned on the same OpenThoughts dataset. **Only if the pruned performance is better than that model, we can know the experiment the authors presented at least makes sense instead of just some random numbers or wrong results**
  - Still, the above experiment could be less accurate, since the referece model is without pre-training. A fairer comparison could be between the pruned 8B→5B model (≈60% pruning) and an existing 3B–4B model (e.g., LLaMA-3.2-3B) fine-tuned on OpenThoughts.  The pruned results should be at least comparable to it, then we can say the experiment is reliable. Without this reference, the pruning results are meaningless.
- No Technical or Methodological Contribution. The paper merely applies several existing pruning methods to two models and reports the numbers. There is **no new pruning algorithm, no new adaptation for reasoning models, and no meaningful theoretical or empirical insight**. Simply reusing prior methods without a solid scientific question or new methodological contribution **does not meet the bar for ICLR**. Combining weakness 1 and 2, I do not think this paper meets the standard for a publishable paper at ICLR.

- The correct way to run your experiments. If you cannt get a reasonable pruning results, then that means the way you use those methods are incorrect. The standard practice of pruning [1-4] is:

  - **Prune during or after pre-training**, not after fine-tuning.

  - **Continue pre-training and fine-tuning** on the pruned model to recover performance and deliver the final model.

  Applying pruning after SFT, especially on thinking models, will of course hurt performance. You may ask why in your experiments the evaluation results of `Llama 3.1-Tulu-3-8B-SFT` after pruning is still not that bad. I think this is because the evaluation tasks are all pre-training evaluation tasks T If you want to the pruning correctly, the setup would have been:

  - Instruct models. Given a pre-trained model (without SFT), prune it with existing approaches [1,2,3,4]. After you get the pruned model (which is stronger than small-scale models pre-trained from scratch with the same number of activated parameters as the prune model), perform SFT on it and deliter the final model. You can also refer to [3] for Apple's practice for developing the on-device model via pruning and continued pretraining + post-training and [4, 8] for NVIDIA's thinking models which are first pruned from a larger models followed by continued pretraining + reasoning post-training.
  - Thinking models. Similarly, you shoud first prune the pre-trained models; after you prune, you fine-tune it on OpenThoughts as the performance of pruning -- it is not the way you used, where you first fine-tuen then prune.

- Unsupported claims. Footnote 2 (page 5) states:

  > “To our knowledge, it is the first reasoning model trained on a fully open corpus, with both the model and its training data publicly released.”

  This claim is **false**. Multiple fully open reasoning models and datasets predate this work, including OpenMathReasoning [5], AM-Thinking-v1 [6], and OpenReasoning-Nemotron series [7]. Making such an assertive statement without checking literature **shows poor scholarship and lack of familiarity with the field**.


Overall, this paper gives the reader the impression that **it originates from a failed project where the authors tried to perform pruning on thinking models rather than a rigorous scientific investigation**. It seems the authors attempted to prune a thinking model, observed performance collapse, and then decided to generalize this failure into a claim that “reasoning models cannot be pruned.”

However, this conclusion is highly likely to be wrong. The industrial efforts [8] have already successfully demonstrated pruning (NVIDIA call their method as neural architecture search, but it is basically searching which components in each layer of the LLM are most important and prune the others) for thinking models when implemented correctly, i.e., by pruning during or after pretraining followed by continued pretraining and post-training. This contrasts with this paper’s “0% performance of pruned models" and strongly suggests that the failure arises from an incorrect pipeline and poor experimental design, not an inherent property of reasoning models.



[1] Xia, Mengzhou, et al. "Sheared LLaMA: Accelerating Language Model Pre-training via Structured Pruning." The Twelfth International Conference on Learning Representations.

[2] Sreenivas, Sharath Turuvekere, et al. "Llm pruning and distillation in practice: The minitron approach." arXiv preprint arXiv:2408.11796 (2024).

[3] Gunter, Tom, et al. "Apple intelligence foundation language models." arXiv preprint arXiv:2407.21075 (2024).

[4] Bercovich, Akhiad, et al. "Puzzle: Distillation-Based NAS for Inference-Optimized LLMs." Forty-second International Conference on Machine Learning, 2025.

[5] Moshkov, Ivan, et al. "Aimo-2 winning solution: Building state-of-the-art mathematical reasoning models with openmathreasoning dataset." arXiv preprint arXiv:2504.16891 (2025).

[6] Ji, Yunjie, et al. "AM-Thinking-v1: Advancing the Frontier of Reasoning at 32B Scale." arXiv preprint arXiv:2505.08311 (2025).

[7] https://huggingface.co/collections/nvidia/openreasoning-nemotron and https://huggingface.co/datasets/nvidia/Nemotron-Post-Training-Dataset-v1

[8] Bercovich, Akhiad, et al. "Llama-nemotron: Efficient reasoning models." arXiv preprint arXiv:2505.00949 (2025).

**Questions:**

Please refer to the weakness above

---

### Official Review · Reviewer_Bs2H · 2025-11-01

**Soundness:** 2
**Presentation:** 2
**Contribution:** 2
**Rating:** 4
**Confidence:** 4

**Summary:**

The paper introduces the first systematic study of pruning for both instruction-following (LLM-Instruct) and reasoning-augmented (LLM-Think) language models. Besides, the paper proposes a unified experimental framework that conducts pruning calibration and post-fine-tuning. Through extensive experiments on 17 benchmarks comparing three structured pruning strategies: static depth, static width, and dynamic depth pruning, the experimental results demonstrate the pruning methods tailored for concise instruction-following do not generalize well to maintaining complex, multi-step reasoning capabilities.

**Strengths:**

- The paper first comprehensive studies comparing pruning across instruction-following models (LLM-instruct) and reasoning-augmented models (LLM-think).

- The paper presents a comprehensive experimental design, covering 17 diverse datasets and three mainstream pruning strategies.

**Weaknesses:**

- The pruning methods used in Table 1 are relatively limited, particularly for Static Depth Pruning and Static Width Pruning. The paper needs to include more pruning methods such as SLEB[1], PuDDing[2], Blockpruner[3], Olica[4], LoRAP[5] in the experiments.

- The conclusion presented in this section 4.3 appears to have already been discussed  in the paper[6].

- The paper does not propose a new pruning method but rather conducts a systematic comparison of existing approaches. As a result, the contribution is mainly observational and experimental rather than methodological.

  [1] Song J, Oh K, Kim T, et al. SLEB: Streamlining LLMs through Redundancy Verification and Elimination of Transformer Blocks[C]//Forty-first International Conference on Machine Learning.

  [2] Wee J, Park M, Lee J. Prompt-based Depth Pruning of Large Language Models[C]//Forty-second International Conference on Machine Learning.

  [3] Zhong L, Wan F, Chen R, et al. Blockpruner: Fine-grained pruning for large language models[J]. arXiv preprint arXiv:2406.10594, 2024.

  [4] He J, Lin H. Olica: Efficient Structured Pruning of Large Language Models without Retraining[C]//Forty-second International Conference on Machine Learning.

  [5] Li G, Tang Y, Zhang W. LoRAP: Transformer Sub-Layers Deserve Differentiated Structured Compression for Large Language Models[C]//Forty-first International Conference on Machine Learning.

  [6] Ji Y, Xiang Y, Li J, et al. Beware of Calibration Data for Pruning Large Language Models[C]//The Thirteenth International Conference on Learning Representations.

**Questions:**

Please refer the Weaknesses.

---

> ### Author Response · Authors · 2025-12-02
>
> 1. **The pruning methods used in Table 1 are relatively limited, particularly for Static Depth Pruning and Static Width Pruning. The paper needs to include more pruning methods such as SLEB, PuDDing, Blockpruner, Olica, LoRAP in the experiments.**
>
>     Thank you for the suggestion. In the revised version, we have **added several additional static pruning baselines** to strengthen the comparison. Beyond the original methods reported in Table 1, we now include **SLEB and BlockPruner**. These methods cover a broader range of static pruning strategies, enabling a more comprehensive evaluation of our approach.
>
> 2. **The presentation in section 4.3 appears to have already been discussed in the paper.**
>
>      Thank you for the comment. Our work specifically examines whether using **the model’s original training corpus**, both as the calibration set and the post-training dataset, can better preserve and recover performance after pruning. This aspect was not explored in prior sections. By aligning pruning with the model’s native data distribution, we find that the pruned models retain their capabilities more reliably. Thus, Section 4.3 provides new insights beyond standard calibration analyses and highlights the importance of distribution-aligned post-training for pruned LLMs.

---

### Official Review · Reviewer_fhQY · 2025-11-07

**Soundness:** 1
**Presentation:** 2
**Contribution:** 2
**Rating:** 2
**Confidence:** 3

**Summary:**

The paper highlights that most structured-pruning studies focus on instruction-tuned models and investigates whether those pruning strategies transfer to reasoning-centric models. It compares static depth, static width, and dynamic depth pruning on two controlled models (LLM-instruct vs. LLM-think) across 17 classification, generation, and reasoning benchmarks, and further studies the effect of calibration data on pruning and post-pruning recovery using the models’ original training corpora. In conclusion, the paper argues that (1) pruning strategies that are optimal for LLM-instruct do not necessarily transfer to LLM-think, and (2) while using the original training corpus during pruning itself yields limited impact, employing it during the recovery phase can lead to non-neligible effect on performance of downstream tasks.

**Strengths:**

- The paper presents a well-motivated study that clearly identifies a neglected gap in prior pruning research, the mismatch between instruction-tuned and reasoning-centric models, and frames this as a timely and practically relevant problem for the community.
- The paper provides a set of elementary yet informative experiments that reveal specific failure modes when structured pruning is applied to reasoning-centric models

**Weaknesses:**

- Although the paper positions itself as a systematic study, both the breadth and depth of the experiments appear insufficient to fully support this claim.
    - In terms of breadth, the study evaluates pruning behavior on only a single, relatively small-scale model, which limits the generalizability of its conclusions and weakens the argument that its findings will reliably transfer to subsequent work.
    - In terms of depth, the paper offers little to no mechanistic analysis of why reasoning-centric models exhibit higher pruning sensitivity than instruction-tuned models, which limits the actionable insights to the community.
- While the paper frames its OpenThoughts SFT-tuned LLaMA-3.1 8B model as a reasoning-centric LLM, this model may not faithfully reflect the behavior of contemporary reasoning models, which rely on specialized post-training pipelines such as RL-based reasoning optimization (e.g., DeepSeek-R1, o1)

**Questions:**

- In line 453, the authors state that ShortGPT is selected to measure the effect of calibration. However, given that each pruning method operates under substantially different mechanisms, it is unclear whether insights from a single method can be generalized to characterize the role of calibration in structured pruning as a whole. The effect of calibration may not be consistent across pruning strategies.
    - Could the authors provide calibration results across additional structured pruning approaches to verify whether the trends are method-agnostic or method-dependent?

- It is not fully clear to me why it is crucial to examine the difference in pruning behavior between LLM-instruct and LLM-think specifically within the proposed framework (where we can utilize the original training corpus).
    - Could the authors elaborate on why their particular experimental framework is the necessary or most appropriate lens for doing so?

- I wonder if the authors performed any qualitative analysis of the reasoning traces produced by the pruned reasoning models. In particular, did they observe characteristic failure patterns such as token repetition or something?

---

> ### Author Response · Authors · 2025-11-30
>
> 1. **Could the authors provide calibration results across additional structured pruning approaches to verify whether the trends are method-agnostic or method-dependent?**
>
> Thanks for your comments. We conducted additional calibration experiments using the static width pruning method SliceGPT to evaluate whether the observed trends hold across structured pruning methods. For static depth pruning, as in ShortGPT, calibration has only a minor effect, while post-training accounts for most of the performance recovery. In contrast, for SliceGPT, both calibration and post-training are important, with the best results achieved when both are aligned with the model’s original training corpus. Although the absolute gains vary due to differences in pruning methods, the overall patterns—namely, the benefits of using the original training corpus—remain consistent. These findings suggest that the conclusions from our main experiments generalize across structured pruning strategies and that leveraging the model’s original training corpus for pruning calibration further helps preserve and recover the pruned model’s performance.
>
> 2. **Could the authors elaborate on why their particular experimental framework is the necessary or most appropriate lens for doing so?**
>
> Thanks for your comments. Our framework leverages the model’s **original training corpus** for both post-training and calibration, providing a controlled and consistent setting for evaluating pruning behavior. This is crucial for comparing LLM-instruct and LLM-think, as these models encode different reasoning patterns and token-level dynamics. Using the native corpus allows us to **isolate the effect of pruning itself** and to apply post-training and calibration in a way that **better preserves and recovers the pruned model’s performance**, avoiding confounding factors such as distribution shift or mismatched data. As a result, we can directly examine how pruning alters block importance and layer utilization, ensuring that the observed differences reflect the models’ **intrinsic pruning sensitivities** rather than artifacts introduced by external datasets.
>
> 3. **I wonder if the authors performed any qualitative analysis of the reasoning traces produced by the pruned reasoning models. In particular, did they observe characteristic failure patterns such as token repetition or something?**
>
> Thank you for the suggestion. We have conducted qualitative analyses of the reasoning traces produced by the pruned reasoning models. Consistent with the reviewer’s intuition, we do observe characteristic failure patterns: models that lose too much reasoning capacity tend to produce **repetitive or looping outputs**, often indicating that the pruning has disrupted key intermediate reasoning steps.

---

### Official Review · Reviewer_ft7S · 2025-11-08

**Soundness:** 1
**Presentation:** 2
**Contribution:** 1
**Rating:** 2
**Confidence:** 4

**Summary:**

This paper addresses the limitation that most pruning algorithms evaluate model performance on calibration sets that differ from the model’s original training data, leaving it unclear whether the model’s native general capabilities are truly preserved. To ensure a faithful assessment, the authors use models whose training datasets are publicly available—both instruction-tuned (LLM-instruct) and reasoning-augmented (LLM-think) models—and perform pruning using the same datasets on which the models were originally trained. They further argue that these different model types require distinct pruning strategies. Accordingly, the paper compares static depth pruning, static width pruning, and dynamic depth pruning across classification \& generation tasks in LLM instruct model, and reasoning tasks in LLM reasoning model. The authors claim that while dynamic depth pruning is effective for classification and generation tasks, static width pruning demonstrates greater robustness on reasoning tasks.

**Strengths:**

1. The motivations are clearly presented.

2. The related works and preliminaries provide a fair and comprehensive coverage of the relevant literature.

2. The pruning strategies are well categorized into three main types—static depth pruning, static width pruning, and dynamic depth pruning—each encompassing recently published algorithms.

**Weaknesses:**

1. Weak Contribution.

This paper applies existing LLM structured pruning methods to LLM-reasoning models. For contribution, the authors should provide additional evidence as follows.

- Does LLM-reasoning models exhibit different trends from LLM-instruct models when using pruning algorithms?
  - For static depth pruning (Shortened LLaMA or SLEB), does the cosine similarity (block importance) differ between the instruct and reasoning models?
  - For static width pruning (LLM-Pruner, SliceGPT), do the pruned heads (in MHA) or channels (in FFN) differ?
  - For dynamic depth pruning (MoD, D-LLM), do the pruned tokens differ?

2. Weak Novelty.

More insights or explanations should be provided if LLM-reasoning models are too sensitive in existing pruning algorithms.
  -  When pruning reasoning models and evaluating them on classification and generation tasks, are they more sensitive than instruct models?
  - For pruning reasoning models, what specific challenges must be addressed in pruning algorithms to prevent catastrophic failure?

3. Weak Faithful assessment.

The authors correctly point out that using only the C4 dataset for pruning can limit domain diversity, and I agree with this concern. However, it remains unclear whether their proposed approach—using the model’s original training dataset for calibration—effectively resolves this issue. Prior works [1], [2] have shown that using calibration sets covering multiple domains can lead to more consistent pruning performance, suggesting that domain diversity plays a crucial role in evaluating pruning robustness.

While this setup aims to ensure faithfulness by aligning the calibration and training distributions, the paper does not provide evidence that the training dataset itself adequately covers diverse domains, nor that the sampling process used to construct the calibration subset avoids distribution shift. Without such analysis, it is difficult to conclude whether the model’s native general capabilities are truly preserved.

Providing quantitative analysis or visualization showing that the calibration data preserve the diversity and statistical characteristics of the full training dataset would help demonstrate that the evaluation faithfully reflects the model’s general capabilities.

4. Weak Justification of Conclusions and Missing Analysis.

The authors conclude that static width pruning shows the greatest robustness, but the evidence supporting this claim is limited and unconvincing. In Section 4.1, only SliceGPT appears robust, while LLM-Pruner’s performance is unstable. Moreover, the reported averages (e.g., 18.50 vs. 11.27) occur only in MATH or GPQA, whereas most other reasoning tasks collapse to near-zero performance, making the conclusion about robustness questionable.

- Could the authors provide stronger empirical or analytical justification for why static width pruning is considered the most robust strategy across reasoning tasks?

The paper lacks analysis on the failure of dynamic depth pruning.

- Why does the reasoning task completely collapse in dynamic depth pruning? A deeper investigation into this behavior or providing theoretical limitation of this algorithm would strengthen the contribution.

[1] Bandari et al., "Is C4 Dataset Optimal for Pruning? An Investigation of Calibration Data for LLM Pruning", EMNLP, 2024.

[2] Williams et al., "On the Impact of Calibration Data in Post-training Quantization and Pruning", ACL, 2024

**Questions:**

Questions are already included in the Weakness section. Additional questions are as follows:

1. Are there any pruning experiments conducted on other open-source datasets using both LLM-instruct and LLM-reasoning models? If so, do they show consistent results with those observed on the LLaMA family?

2. Please specify the exact hyperparameter settings for each pruning algorithm (e.g., the size of the calibration set, the training cost for router optimization in dynamic depth pruning), and clarify how the experimental conditions were made fair for comparison across methods.

---

> ### Author Response · Authors · 2025-12-02
>
> Thank you for your comments. We compare pruning trends between LLM-instruct and LLM-reasoning models, analyze why LLM-think models are more sensitive, and discuss the challenges of preserving their performance. We also provide quantitative evaluations and visualizations demonstrating that calibrating the model and post-training with its original training data helps maintain its capabilities. All hyperparameters are reported to ensure fair comparisons across methods.

---

### Note · Authors · 2025-12-02

I have read and agree with the venue's withdrawal policy on behalf of myself and my co-authors.